# Polyphenolic Profile, Anti-Inflammatory and Anti-Nociceptive Activities of Some African Medicinal Plants

**DOI:** 10.3390/plants11101377

**Published:** 2022-05-22

**Authors:** Windmi Kagambega, Hadidjatou Belem, Roland Nâg-Tiéro Meda, Benjamin Kouliga Koama, Anne-Flora Drabo, Jacques Kabore, Amadou Traore, Georges Anicet Ouédraogo, Daniela Benedec, Daniela Hanganu, Laurian Vlase, Ana-Maria Vlase, Oliviu Voștinaru, Cristina Mogoșan, Ilioara Oniga

**Affiliations:** 1Laboratoire de Recherche et d’Enseignement en Santé et Biotechnologies Animales, Unité De Formation et de Recherche en Sciences et Techniques, Université Nazi Boni, Bobo-Dioulasso 01 BP 1091, Burkina Faso; kagambegawindmi@yahoo.fr (W.K.); kadibelem1@gmail.com (H.B.); meda_roland@yahoo.fr (R.N.-T.M.); koama290@yahoo.fr (B.K.K.); dr.floranna@yahoo.com (A.-F.D.); ogeorgesanicet@yahoo.fr (G.A.O.); 2Unité de Formation et de Recherche en Sciences et Techniques, Université Nazi Boni, Bobo-Dioulasso 01 BP 1091, Burkina Faso; jacqueskabore@yahoo.fr; 3Laboratoire de Médicine et Pharmacopée Traditionnelle, Institut de Recherche en Sciences de la Santé (IRSS), Bobo-Dioulasso 01 BP 2779, Burkina Faso; 4Centre International de Recherche-Développement sur l’Élevage en Zones Subhumides (CIRDES), Unité de Recherches sur les Bases Biologiques de la Lutte Intégrée, Bobo-Dioulasso 01 BP 454, Burkina Faso; 5Laboratoire de Biologie et Santé Animales, Institut de l’Environnement et de Recherches Agricoles (INERA), Ouagadougou 04 BP 8645, Burkina Faso; traore_pa@yahoo.fr; 6Faculty of Pharmacy, “Iuliu Hatieganu” University of Medicine and Pharmacy, 8 V. Babes Street, 400010 Cluj-Napoca, Romania; dbenedec@umfcluj.ro (D.B.); laurian.vlase@umfcluj.ro (L.V.); anamaria.gheldiu@yahoo.com (A.-M.V.); oliviu_vostinaru@yahoo.com (O.V.); cmogosan@umfcluj.ro (C.M.); ioniga@umfcluj.ro (I.O.)

**Keywords:** *Parkia biglobosa*, *Detarium microcarpum*, *Vitellaria paradoxa*, *Sclerocarya birrea*, polyphenols, anti-inflammatory, anti-nociceptive

## Abstract

The aim of the present study was to investigate the polyphenolic profile and the anti-inflammatory and anti-nociceptive activities of four traditionally used medicinal plants from Burkina Faso: *Parkia biglobosa*, *Detarium microcarpum*, *Vitellaria paradoxa* and *Sclerocarya birrea*. The analysis of the main phenolic compounds was performed by the HPLC-UV-MS method. The anti-inflammatory effect of the aqueous bark extracts was investigated by the λ-carrageenan-induced rat paw edema test. The anti-nociceptive activity was evaluated by the Randall–Selitto test under inflammatory conditions. Seven phenolic acids (gallic, protocatechuic, gentisic, vanillic, *p*-coumaric, ferulic, and syringic acids), and three flavonoids (catechin, epicatechin, and quercitrin) were identified in the plant samples. High contents of gallic acid were determined in the *D. microcarpum*, *P. biglobosa* and *S. birrea* extracts (190–300 mg/100 g), and *V. paradoxa* extract was the richest in epicatechin (173.86 mg/100 g). The *λ*-carrageenan-induced inflammation was significantly reduced (*p* < 0.001) by the *P. biglobosa* and *D. microcarpum* extracts (400 mg/kg p.o.). Under the inflammatory conditions, a significant anti-nociceptive activity (*p* < 0.001) was obtained after 2–3 h from the induction of inflammation. The effects of the tested extracts could be related to the presence of polyphenols and could be useful in the management of certain inflammatory diseases.

## 1. Introduction

About 80% of the world’s population relies on medicinal plants for primary health care needs [1,2]. The African continent is endowed with an enormous wealth of plant resources, but only 25% of known species are used as medicine remedies for the prevention and treatment of different diseases. Africa remains a minor player in the global natural products market, largely due to a lack of scientific information regarding natural products [1,2]. 

Further, despite deep concerns about the mechanisms of action for the bioactive components of some plants belonging to the African flora, there is still a lack of up-to-date literature. The procedures adopted to ensure the quality, authenticity and standardization of raw vegetable products must be improved and followed.

Appropriate quality could be achieved through the proper management of raw material, extraction procedures, and final product formulation [2]. Numerous ethnobotanical studies show the various potentials of plants without presenting scientific evidence for their traditional use. Moreover, there are many situations in which potentially medicinal plants are under-studied, as in the case of the species that represent the subject of our study: *Parkia biglobosa* (Jacq.) R. Br. ex G. Don, also named ‘African locust bean’; *Detarium microcarpum* Guill. & Perr., with the common name ‘sweet dattock’; *Vitellaria paradoxa* C. F. Gaertn or ‘shea butter tree’; and *Sclerocarya birrea* (A. Rich.) Hochst, named ‘marula’. Most studies that have been carried out on these species provide argument for the scientific basis of the therapeutic use of these plants [3,4,5]. 

Medicinal plants exert favorable effects on animal and human health due to their content of secondary metabolites, such as polyphenols [6,7]. Polyphenols have been reported to have anti-inflammatory, antimicrobial, antioxidant, immunomodulatory, anti-mutagenic, and anti-allergic properties [8,9]. They can alleviate infection-related inflammatory diseases via the modulation of the pathogen recognition receptor-—mediated signaling pathways [10]. As such, this study was undertaken to complement the available scientific data for the development of qualitative natural remedies in order to valorize African flora resources.

The aim of this research was to select some medicinal plants that are traditionally used as anti-inflammatory agents, in order to be included in nutraceutical formulations. In a previous study, total polyphenols and flavonoids contents, together with antioxidant activity were determined for the analyzed species [11]. The present study identified and quantified important phenolic compounds that were found in extracts of important African medicinal plants. Furthermore, these extracts were evaluated for their anti-inflammatory and antinociceptive effects. Our results indicate that these plant extracts have the potential to be effective anti-inflammatory agents.

## 2. Results

### 2.1. HPLC-UV-MS Analysis

A HPLC-UV-MS analysis of the four plants’ aqueous extracts revealed the presence of several polyphenols, both phenolic acids and flavonoids (Table 1). The identified compounds were: six phenolic acids (gallic, protocatechuic, vanillic, syringic, *p*-coumaric, and ferulic acids); two flavanols (catechin and epicatechin); and one flavonoid glycoside (quercitrin = quercetin 3-O-rhamnoside). Gallic acid, protocatechuic acid, vanillic acid, syringic acid, catechin and epicatechin were identified in all plant samples. *p*-Coumaric acid was quantified only in the extract of *P. biglobosa* and gentisic acid was under the limit of quantification in all samples. Gallic acid was the major compound in three extracts: *S. birrea* (299 mg/100 g extract); *P. biglobosa* (222 mg/100 g extract); and *D. microcarpum* (191 mg/100 g extract). In *V. paradoxa* extract the major compound was epicatechin (173 mg/100 g extract) with a low amount of gallic acid (22 mg/100 g extract), compared to other extracts. Catechin was identified in high concentrations in *D. microcarpum* (54 mg/100 g extract) and *V. paradoxa* (37 mg/100 g extract) extracts.

### 2.2. Anti-Inflammatory Activity 

The studied aqueous extracts and diclofenac (reference drug) were tested for their in vivo anti-inflammatory activity by modified carrageenan-induced rat paw edema test [12]. The results are presented in Table 2. An intraplantar injection of 0.1 mL of 1% carrageenan in isotonic saline solution was administered in the left hind paw of the rat-induced edema which progressively increased with a maximum noted at three hours (2.766 mL ± 0.043). The groups that were treated orally with the aqueous extracts showed an inhibition of edema. Thus, after one hour, the extracts of *P. biglobosa* and *D. microcarpum* reduced the edema of the left hind paw with 42.5 and 21.82%, respectively (*p* < 0.01). *V. paradoxa* and *S. birrea* extracts started to act significantly from the third hour with inhibition percentages of 24 and 12.29% (*p* < 0.01), respectively. The administration of a reference anti-inflammatory drug, diclofenac 20 mg/kg p.o., inhibited the formation of edema in treated rats with the effect peaking at 3 h and decreasing slightly thereafter.

### 2.3. Antinociceptive Activity

The antinociceptive activity of the extracts in an inflammatory pain model was evaluated by the Randall–Selitto test [12,13]. The results are presented in Table 3. The maximum weight that was applied on the paw of negative control rats until a response was obtained was equal to 96 ± 2.92 g. These results indicated that after two and three hours from inducing inflammation, the weights applied to the paws of the animal group that received the aqueous extracts (400 mg/kg b.w.) and diclofenac (20 mg/kg b.w.) were significantly higher (*** *p* < 0.001), compared to the negative control group. No significant difference was found between the extracts and the negative control at one hour after the induction inflammation. Only *P. biglobosa* and *D. microcarpum* extracts remained effective after the fourth hour. When comparing the different groups that received the extracts, the weight applied to the group that received *P. biglobosa* extract (400 mg/kg b.w.) was significantly higher (*p* < 0.01) than that of the groups that received the other extracts.

## 3. Discussion

In this study, the bark aqueous extracts of four African plant species were analyzed: *S. birrea*, *V. paradoxa*, *P. biglobosa*, and *D. microcarpum.* These species are known for the significant nutritional values of their fruits, but the medicinal potential of the bark is less studied. The use of these plants in folk medicine is the starting point for obtaining good quality, effective and safe medicinal products for the treatment and prevention of different diseases [11]. 

A HPLC-MS analysis of the aqueous extracts of the studied species revealed the presence of several phenolic acids (gallic, protocatechuic, vanillic and syringic acids), as well as flavonoids (catechin and epicatechin) with a varied distribution in the analyzed samples. *S. birrea* bark extract was the highest in phenolic acids, especially gallic acid, while *V. paradoxa* extract contained the highest quantity of flavonoids, especially epicatechin. Phenolic acids were better represented in *P. biglobosa* bark extract comparing to flavonoids, while *D. microcarpum* extract contained intermediate concentrations of phenolic acids and flavonoids.

The studied plant extracts demonstrated potential in vivo anti-inflammatory and analgesic activity. In effect, the oral treatment of rats with the aqueous extracts of *P. biglobosa* and *D. microcarpum* reduced the carrageenan-induced edema at 1 h which was superior to the positive control. The best anti-inflammatory effect was observed for *P. biglobosa* at all four measurement moments, compared to the positive control (diclofenac). In addition, these extracts showed antinociceptive properties at the second and third hour after the induction of inflammation. Our results are similar to those reported by Yaro et al., [14] that indicated significant analgesic and anti-inflammatory properties for *D. microcarpum* stem-bark methanol extract. These activities could be due to the presence of polyphenolic molecules, the analyzed extract being rich in these compounds. Indeed, previous studies demonstrated the role of gallic acid as an anti-inflammatory, antioxidant, anti-anaphylaxis, anti-tumor, anti-radiation agent [15,16,17,18,19,20]. Thus, gallic acid can modulate pro-inflammatory gene expression and cytokine production. The in vivo experiments showed that extracts containing gallic acid together with other polyphenols had a clear restrictive effect on local inflammation in the animal model (*p* < 0.05) [21]. 

In pain control systems, the integration of the supraspinal/spinal signals mechanism includes the release of several neurotransmitters, particularly the endogenous opioids, norepinephrine, serotonin and acetylcholine [22]. Thus, the involvement of these modulatory systems in protocatechuic acid-induced analgesia was investigated, and the results indicated that protocatechuic acid had an antinociceptive action [23]. Analgesic and anti-inflammatory effects were shown also in vanillic acid [24,25].

Among the most frequently reported pathologies, colibacillosis or infection due to pathogenic *Escherichia coli* are responsible for the inflammation of multiple organs [26]. Indeed, this pathogenic bacterium causes massive production of inflammatory cytokines and adhesion molecules via the activation of Toll-like receptors (TLRs) to the NF-κβ (TLRs/NF-κβ) signaling pathway [27]. Although TLR-mediated signaling is fundamental to fighting infections and other diseases and enhancing tissue repair, the regulation of these signals must be controlled [26]. Stimulation of the TLR signaling pathway plays a critical role in triggering innate immune responses and defending against microbial infections [28]. However, upregulation of these receptors can disrupt immune balance and lead to a continuous inflammatory state and poor signaling to the adaptive system [29]. Polyphenols in general can mitigate inflammatory diseases related to infections via the modulation of the signaling pathways that are mediated by pathogen recognition receptors [30,31]. Thus, polyphenols were found to effectively regulate NF-κβ activation and reverse TLR gene over-expression in exposed organisms [10].

The anti-inflammatory and antinociceptive effects of *P. biglobosa* and *D. microcarpum* aqueous extracts which contain polyphenolic compounds such as gallic, protocatechuic and vanillic acids suggests that they are promising herbal drugs for pain and inflammation management. A good antioxidant activity that was previously tested [11] is added to these properties. However, further tests are needed to elucidate the mechanisms involved in the anti-inflammatory and analgesic activity, as these effects could also be related to other non-phenolic molecules.

## 4. Materials and Methods

### 4.1. Plant Materials and Chemicals 

The raw material was represented by the barks of 4 species of plants harvested in April 2019 from the Bobo-Dioulasso region, Burkina Faso (11°13′21″ north, 4°25′37″ west). The species were identified by Doctor of Botany and cytotechnologist, Yempabou Hermann Ouoba, from the Nazi Boni University of Bobo, Dioulasso (UNB). The collected bark was dried at room temperature and then powdered using a Retsch Knife Mill Grindomix GM 200 (RETSCH GmbH, Eragny, France). The studied species were *Parkia biglobosa* (Jacq.) R.Br. ex G.Don (Fabaceae/Mimosaceae); *Detarium microcarpum* Guill. & Perr. (Fabaceae); *Vitellaria paradoxa* C.F.Gaertn. (Sapotaceae); and *Sclerocarya birrea* (A. Rich.) Hochst (Anacardiaceae).

Some HPLC references, such as chlorogenic acid, *p*-coumaric acid, caffeic acid, rosmarinic acid, rutin, apigenin, quercetin, isoquercitrin, quercitrin, hyperoside, kaempferol, myricetin, and fisetin were purchased from Sigma Aldrich (St. Louis, MO, USA). Cichoric acid, caftaric acid, ferulic acid, sinapic acid, gentisic acid, gallic acid, patuletin, and luteolin were obtained from Roth (Karlsruhe, Germany). HPLC grade solvents were purchased from Merck (Darmstadt, Germany). All chemicals and reagents were of high-grade purity.

λ-carrageenan and diclofenac were purchased from Sigma Aldrich, St. Louis, MO, USA.

### 4.2. Preparation of Extracts

The extracts were prepared by heating a mixture of 50 g of vegetable powder and 500 mL of distilled water at 100 °C under reflux for 30 min. The obtained extracts were subsequently lyophilized using a lab scale VirTis Advantage Plus freeze-drier (SP Scientific, Gardiner, ME, USA) [11]. The yields of each extract were 10.61% ± 0.10 for *Parkia biglobosa*, 8.26% ± 0.15 for *Sclerocarya birrea*, 10.14% ± 0.24 for *Detarium microcarpum* and 12.20% ± 0.31 for *Vitellaria paradoxa*. 

### 4.3. HPLC-UV-MS Analysis—Apparatus, Chromatographic Conditions and Polyphenols Determination

An Agilent 1100 HPLC Series system (Agilent, Santa Clara, CA, USA) coupled with an Agilent 1100 mass spectrometer (LC/MSD Ion Trap SL) was used for the identification and quantification of the polyphenols (caftaric acid, gentisic acid, caffeic acid, chlorogenic acid, *p*-coumaric acid, ferulic acid, sinapic acid, hyperoside, isoquercitrin, rutin, myricetin, fisetin, quercitrin, quercetin, patuletin, luteolin, kaempferol and apigenin as references). The compounds were separated on a reverse-phase analytical column (Zorbax SB-C18 100 × 3.0 mm i.d., 3.5 μm particle) and the polyphenols were identified on both UV mode (330–370 nm) and MS (electrospray ion source in negative mode; capillary +3000 V, nebulizer 60 psi (nitrogen), dry gas nitrogen at 12 mL/min and dry gas temperature 360 °C). A binary gradient that was prepared from methanol and a solution of acetic acid 0.1% (*v*/*v*) was used as mobile phase, starting with 5% methanol and ending at 42% methanol at 35 min. For the next 3 min, isocratic elution was used with 42% methanol. The column was rebalanced in the next 7 min with 5% methanol. The column temperature was set at 48 °C. ChemStation and DataAnalysis software (from Agilent, USA) were used for processing the obtained data [32,33]. The compounds were identified by comparison of their retention times (Rt) and the MS spectra with those of the corresponding references. An external standard method was used for the quantification of the polyphenols based on calibration curves in the 0.5–50 μg/mL range with good linearity (R^2^ > 0.999) [32,33].

In another stage of work, 6 polyphenols (epicatechin, catechin, syringic acid, gallic acid, protocatechuic acid and vanillic acid) were evaluated by a different HPLC-MS method. The same column was used for separation (Zorbax SB-C18) and the mobile phase was made by methanol (M) and 0.1% acetic acid (A, *v*/*v*) in a binary gradient (0 min: 3% M; 0–3 min: 8% M; 3–8.5 min: 20% M; 8.5–10 min 20% M and finally 3% M, for a rebalance of the column). The compounds were identified on MS mode (SIM-MS) with an electrospray ion source in negative mode and by comparison of their Rt with references [34]. 

The injection volume was 5 µL and the flow rate was set at 1 mL/min for both chromatographic methods. All determinations were conducted in triplicate.

### 4.4. Evaluation of Anti-Inflammatory and Anti-Nociceptive Activities

#### 4.4.1. Animals

Six groups of female Charles River Wistar rats (n = 5) that were between 230 and 370 g in weight were used to carry out the pharmacological activities. The animals were provided by the Center for Practical Skills and Experimental Medicine of the “Iuliu Hatieganu” University of Medicine and Pharmacy (Cluj-Napoca, Romania). The animals were housed in open-top type IV-S polycarbonate cages (Tecniplast, Italy) and kept at a controlled room temperature (22 ± 2 °C; relative humidity 45 ± 10%) with a light/dark cycle of 12/12 h. They were fed with a standard diet of pellets and had access to water *ad libitum* [12]. Prior to any experimentation, the animals were left in a fasting state for 12 h. The procedures on the animals were carried out in accordance with the EEC Directive 63/2010 which regulates the care and use of laboratory animals for scientific purposes and approved by the Veterinary Health and Food Safety Directorate of Cluj-Napoca (Authorization no.238/24 December 2020). 

#### 4.4.2. Anti-Inflammatory Activity 

The anti-inflammatory effect of the extracts was determined by the rat paw edema test induced by λ-carrageenan [12]. The acute inflammation was induced 1 h after intragastric administration of the substances by an intraplantar injection of 100 μL of λ-carrageenan 1% saline solution into the left hind paw of the rats. The volume (mL) was measured by the plethysmometer (Ugo Basile 7140, Varese, Italy) at 0, 1, 2, 3 and 4 h after injection of λ-carrageenan. The volume of edema and the percentage of edema inhibition were expressed as follows:
Volume edema = V_t_ − V_0_ (mL)
Inhibition of edema = [1 − E_t_/E_c_] × 100, 
where: V_0_ is the mean edema volume before injection of λ-carrageenan; V_t_ is the mean edema volume at time t after injection of λ-carrageenan; E_t_ is the mean edema volume of treated animals; and E_c_ is the mean edema volume of animals in the negative control group [12]. 

#### 4.4.3. Anti-Nociceptive Activity

The anti-nociceptive activity of the extracts was evaluated by the Randall–Selitto test under inflammatory conditions [13]. This test was performed simultaneously with the λ-carrageenan-induced rat paw edema test and evaluates the nociceptive withdrawal threshold. In this model of inflammatory pain, a linearly increased force is applied on the inflamed paw and the pain threshold is determined at 0, 1, 2, 3 and 4 h after the intraplantar injection of λ-carrageenan using the analgesimeter (Ugo Basile 37215, Varese, Italy). The analgesimeter applies a linearly increased force (in grams) until the animal produces a response that is characterized by paw retraction or vocalization interpreted as mechanical hypernociception. The maximum applied weight where the response was obtained was noted at each time interval [12,13].

#### 4.4.4. Treatment

Each extract was administered at a dose of 400 mg/kg body weight. Rats in the negative control group were treated with normal saline solution. Animals in the positive control group received a reference anti-inflammatory/antalgic drug (diclofenac 20 mg/kg of body weight (b.w.)). Intragastric administration of the substances was made 1 h before induction of the inflammation [12].

#### 4.4.5. Statistical Analysis

The statistical analysis was performed with a two-way analysis of variance (ANOVA), followed by Dunnett‘s multi-comparison test. All data are presented as percentage ± standard error of mean, n = 5 rats.

## 5. Conclusions

The present study analyzed the phenolic compounds and evaluated the anti-inflammatory and anti-nociceptive activities of different extracts obtained from four medicinal plants that are native to Burkina Faso. The freeze-dried aqueous extracts were rich in polyphenols, such as phenolic acids (gallic acid, protocatechuic acid, and vanillic acid) and flavonoids (catechin and epicatechin). The extracts of *D. microcarpum* and *P. biglobosa* barks showed the highest anti-inflammatory and anti-nociceptive activities. The main polyphenolic compounds identified in the studied extracts (e.g., gallic acid) could be responsible for the pharmacological activities. The obtained results allowed a better phytochemical and pharmacological characterization of the analyzed species and represent an important part of studies underlining the use of these medicinal plants for health protection.

## Figures and Tables

**Table 1 plants-11-01377-t001:** Phenolic compounds identified in the extracts (mg/100g extract).

Phenolic Compounds	Rt	*m/z*	*P.* *biglobosa*	*D.* *microcarpum*	*V. paradoxa*	*S.* *birrea*
Gallic acid	1.50 ± 0.01	169	222.68 ± 6.12	191.14 ± 3.86	22.82 ± 0.18	299.03 ± 7.47
Protocate- chuic acid	2.80 ± 0.01	153	0.82 ± 0.08	5.86 ± 0.13	3.04 ± 0.09	1.83 ± 0.07
Gentisic acid	3.5 ± 0.01	153	<LOD	<LOQ	<LOQ	<LOQ
Catechin	3.69 ± 0.04	289.2	0.34 ± 0.02	54.82 ± 1.06	37.00 ± 0.78	0.90 ± 0.06
Vanillic acid	5.6 ± 0.04	167	1.76 ± 0.03	2.40 ± 0.07	0.71 ± 0.08	3.16 ± 0.44
Syringic acid	6.0 ± 0.01	197	0.27 ± 0.01	0.93 ± 0.02	0.65 ± 0.09	0.41 ± 0.02
Epicatechin	9.00 ± 0.01	289.2	7.76 ± 0.24	11.85 ± 0.15	173.86 ± 2.14	1.87 ± 0.05
*p*-Coumaric acid	9.48 ± 0.08	163	2.48 ± 0.05	<LOD	<LOQ	<LOD
Ferulic acid	12.8 ± 0.10	193	1.90 ± 0.06	0.30 ± 0.01	<LOD	<LOD
Quercitrin	23.64 ± 0.13	447	<LOD	1.68 ± 0.02	0.84 ± 0.06	<LOD

LOD—limit of detection; LOQ—limit of quantification.

**Table 2 plants-11-01377-t002:** Effect of the extracts on the rat edema induced by λ-carrageenan.

Animal Groups	Dose (mg/kg) p.o.	Edema 1 h (mL) (% of Inhibition)	Edema 2 h (mL) (% of Inhibition)	Edema 3 h (mL) (% of Inhibition)	Edema 4 h (mL) (% of Inhibition)
Negative control	-	1.274 ± 0.046	1.772 ± 0.032	2.766 ± 0.043	2.758 ± 0.018
*P.* *biglobosa*	400	0.734 ± 0.065 *** (42.3%)	1.248 ± 0.039 *** (29.57%)	1.586 ± 0.048 *** (42.66%)	1.748 ± 0.046 *** (36.62%)
*D.* *microcarpum*	400	0.996 ± 0.031 *** (21.82%)	2.126 ± 0.024 (-)	2.442 ± 0.021 *** (11.71%)	2.714 ± 0.048 ns (1.59%)
*V.* *paradoxa*	400	1.206 ± 0.030 ns (5.33)	1.656 ± 0.025 ns (6.54)	2.102 ± 0.021 *** (24%)	2.494 ± 0.027 *** (9.57%)
*S.* *birrea*	400	1.692 ± 0.028 (-)	2.332 ± 0.030 (-)	2.426 ± 0.037 *** (12.29%)	2.578 ± 0.028 * (6.52%)
Diclofenac	20	1.106 ± 0.023 * (13.18%)	1.380 ± 0.023 *** (22.12%)	1.37 ± 0.089 *** (50.46%)	1.544 ± 0.035 *** (44.01%)

* *p* < 0.05; *** *p* < 0.001; ns: no significant difference compared to the control; % inhibition = edema inhibition; p.o.= per os (oral administration).

**Table 3 plants-11-01377-t003:** Nociceptive threshold after oral administration of tested extracts (Randall–Selitto test).

Animal Groups	Dose (mg/kg) p.o.	Linearly Increased Force (g)			
		1 h	2 h	3 h	4 h
**Negative** **control**	-	96.25 ± 2.92	59.40 ± 4.85	54.15 ± 2.92	61.06 ± 2.92
Diclofenac	20	94.50 ± 1.87 ns	100.06 ± 4.00 ***	95.42 ± 2.74 ***	89.30 ± 4.85 ***
*P.* *biglobosa*	400	111.70 ± 4.30 ns	103.76 ± 3.94 ***	101.60 ± 4.00 ***	82.50 ± 4.47 **
*D.* *microcarpum*	400	108.00 ± 4.11 ns	96.60 ± 5.09 ***	95.58 ± 5.70 ***	61.40 ± 2.44 ns
*V.* *paradoxa*	400	101.20 ± 4.30 ns	102.50 ± 3.39 ***	99.60 ± 4.79 ***	73.64 ± 4.63 ns
*S. birrea*	400	88.74 ± 5.78 ns	90.52 ± 5.02 ***	70.22 ± 4.18 *	66.42 ± 4.84 ns

* *p* < 0.05; ** *p* < 0.01; *** *p* < 0.001; ns: no significant difference compared to the control. p.o. = per os (oral administration).

## Data Availability

All results presented in this study were carried out by the authors and the data used as references were properly cited.

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
