# Peer review of "Polyphenolic Profile, Anti-Inflammatory and Anti-Nociceptive Activities of Some African Medicinal Plants"

_plants, 2022, doi:10.3390/plants11101377_

Round 1

Reviewer 1 Report

Attached are comments and suggestions

Reviewer 2 Report

The manuscript intitled “Polyphenolic profile, anti-inflammatory and anti-nociceptive activities of some African medicinal plants” in this work the authors characterized the polyphenolic profile of the aqueous extract of four medicinal plants to then test them for both anti-inflammatory and anti-nociceptive activities. The topic is relevant given that almost 80% of the world population relies on medicinal plants for its primary health care needs.

The manuscript is well written. No grammatical errors were identified. And the structure is clear and concise. The way the manuscript was written is easy to understand. And the presentation of the results is well structured. Furthermore, in the manuscript text, the results are well presented and discussed. The conclusion presents topics consistent with those raised by the hypothesis and with the results obtained. The authors manage to close the cycle presented in the manuscript from beginning to end.
